Subject Areas:
behaviour

Keywords:
cultural evolution, dominance, prestige, social hierarchy, social learning

Author for correspondence:
C. O. Brand
e-mail: c.brand@exeter.ac.uk

# Prestige and dominance-based hierarchies exist in naturally occurring human groups, but are unrelated to task-specific knowledge

## C. O. Brand and A. Mesoudi

Human Behaviour and Cultural Evolution Group, Department of Biosciences, College of Life and Environmental Sciences, University of Exeter, Cornwall Campus, Penryn, UK

 COB, 0000-0003-1285-2174; AM, 0000-0002-7740-1625

Prestige and dominance are thought to be two evolutionarily distinct routes to gaining status and influence in human social hierarchies. Prestige is attained by having specialist knowledge or skills that others wish to learn, whereas dominant individuals use threat or fear to gain influence over others. Previous studies with groups of unacquainted students have found prestige and dominance to be two independent avenues of gaining influence within groups. We tested whether this result extends to naturally occurring social groups. We ran an experiment with 30 groups of people from Cornwall, UK ($n = 150$). Participants answered general knowledge questions individually and as a group, and subsequently nominated a representative to answer bonus questions on behalf of the team. Participants then anonymously rated all other team-mates on scales of prestige, dominance, likeability and influence on the task. Using a model comparison approach with Bayesian multi-level models, we found that prestige and dominance ratings were predicted by influence ratings on the task, replicating previous studies. However, prestige and dominance ratings did not predict who was nominated as team representative. Instead, participants nominated team members with the highest individual quiz scores, despite this information being unavailable to them. Interestingly, team members who were initially rated as being high status in the group, such as a team captain or group administrator, had higher ratings of both dominance and prestige than other group members. In contrast, those who were initially rated as someone from whom others would like to learn had higher prestige, but not higher dominance, supporting the claim that prestige reflects social learning opportunities. Our results suggest that prestige and dominance hierarchies do exist in naturally occurring social groups, but that these hierarchies may be more domain-specific and less flexible than we anticipated.

# 1. Introduction

Prestige and dominance are said to be 'two ways to the top' in gaining status and influence in human social groups [1], and represent two evolutionarily distinct psychological processes found in our species [2]. According to Henrich & Gil-White [2], *dominance* is a form of social influence attained by threat or fear, e.g. of physical aggression or the withholding of resources. By contrast, *prestige* is social influence that is acquired by individuals who are particularly skilled or knowledgeable in a certain domain. Consequently, other individuals in the group voluntarily confer deference on the prestigious individual in order to gain access and proximity to them for social learning opportunities [2]. Over time, prestigious individuals gain status, respect, admiration and attention from other group members who wish to learn from them. While dominance is common across social species, including humans, Henrich & Gil-White [2] argued that prestige is unique to humans due to our unusually extensive reliance on social learning. While terms such as 'prestige' and 'social status' have been defined in many ways across the social sciences [1], Henrich & Gil-White's [2] dominance–prestige scheme usefully captures a key distinction—dominance as acquired coercively due to fear or threat, and prestige as acquired voluntarily due to superior knowledge or skill—within a framework that is evolutionarily grounded and consistent with comparative non-human research.

The prestige–dominance distinction has received much attention across social psychology and evolutionary anthropology since its introduction (for reviews, see [1,3,4]), and has stimulated empirical tests of the many predictions put forth by Henrich & Gil-White [2]. Atkisson *et al.* [5] found that participants in a virtual artefact-design experiment chose to view and copy the artefacts of other group members who had been viewed the most by all other group members. This occurred despite these 'viewing time' attentional cues being fictional, illustrating the potency of prestige-related cues in human social learning. Similarly, Chudek *et al.* [6] found that children copied the food and object choices of adults who had been watched by bystanders, compared to unwatched adults. In a second experiment, Chudek *et al.* found provisional evidence that prestige bias is domain sensitive by showing that children copy the object choices, but not the food choices, of adults previously shown choosing objects but not food (and the reverse for adults shown initially choosing food).

While these studies demonstrate that adults and children use prestige cues (e.g. being attended to by third parties) as guides for choosing from whom to learn, other studies have explicitly compared prestige and dominance as distinct means of attaining status. Cheng *et al.* [7] developed and validated a scale of traits that map strongly onto either prestige or dominance, but not both. Cheng *et al.*'s prestige scale includes items such as 'members of your group respect and admire him/her', and 'his/her unique talents and abilities are recognized by others in the group'. By contrast, the dominance scale includes items such as 'he/she enjoys having control over other members of the group', and 'he/she is willing to use aggressive tactics to get his/her own way'. These combinations of traits have been found to constitute two distinct and viable ways of attaining status and influence over a group [1]. In Cheng *et al.*'s [1] study, previously unacquainted, same-sex student groups completed a decision-making task to rate various items, such as rope, matches and parachutes, in order of importance for use on the moon. Participants rated items individually, then jointly as a group following group discussions, then rated every other group member on prestige and dominance using the aforementioned scales. Prestige and dominance ratings were found to independently predict (i) participants' influence in the task as rated by other group members; (ii) participants' influence in the task as rated by external observers via videotape; and (iii) a behavioural measure of participant influence, specifically the degree to which that participant's individual decision matched the eventual post-discussion group decision. Analyses of the external observers' gaze direction while watching the video footage revealed that individuals with high dominance or prestige ratings also received more visual attention than those with low dominance or prestige ratings.

Our aim here is to extend and verify the findings reviewed above concerning the viability of prestige and dominance as strategies for attaining status within groups. We repeated Cheng *et al.*'s [1] general design in which participants in groups complete a task and rate each other on prestige and dominance using the same scales used by Cheng *et al.* [1,7]. We made two major modifications designed to explore the generality of Cheng *et al.*'s findings.

First, rather than use groups of unacquainted university students brought together for the sole purpose of an experiment, as in Cheng *et al.* [1], we instead recruited already-established community groups of various ages and backgrounds. These included sports clubs, volunteer groups, businesses, bands and chess clubs. This allowed us to test whether dominance and prestige hierarchies are evident in already-established groups of non-students, and whether prestige and dominance affect

decisions within groups who have naturally developed relationships over time. We feel this is an important question because it is rarely the case in reality that individuals make decisions in groups of strangers with whom they may never interact again. Furthermore, our groups also featured a range of ages, social backgrounds and gender compositions, making them more representative of the general population than the same-sex groups of university students used in previous studies [1,7]. This follows recommendations to use broader samples in psychological research [8].

Second, we wanted to test the key prediction of Henrich & Gil-White [2] that prestige is tied to knowledge, while dominance is not. This prediction stems from the assumption that prestige functions to aid individuals in identifying potential demonstrators from whom to learn valuable skills and knowledge. If this is the case, it follows that prestigious individuals should also be more knowledgeable or skilful. Dominant individuals, by contrast, attain their dominance via physical or verbal coercion and threat, and so dominance should be unrelated to knowledge. Cheng *et al.* [1] compared prestige and dominance, but in a task—choosing items to use on the moon—which was not clearly related to a participant's prior or general knowledge. While there were 'correct' answers in the moon task, these were unlikely to depend on participants' past knowledge of moon landings. We therefore used a general knowledge quiz, with a series of multiple-choice questions each of which had a definite correct and incorrect answer, and which relied on participants' existing general knowledge. 'Success' here is therefore the number of correct answers on this quiz task, which is independent of other group members' answers. We then had group members anonymously vote for a single group member to complete a bonus round of quiz questions, with the highest scoring representative winning money for their group. This created an incentive for participants to identify and vote for the most knowledgeable, and therefore most prestigious, group member.

## 1.1. Predictions

All of our predictions were pre-registered and are available at the Open Science Framework link https://osf.io/tasu5/. We recruited 30 groups of mixed-sex, adult participants from already-established community groups across Cornwall, UK, to complete our task. Before the task, we sought to validate the prestige and dominance scales created by Cheng *et al.* [7] by asking participants to choose an influential member of their wider community or country and rate their chosen person on these scales. Based on the results of Cheng *et al.* [1,7] that prestige and dominance are independent means of acquiring status, we made the following prediction:

> H1: Individuals viewed as having high status in the wider community or country are rated as either highly prestigious or highly dominant, but not both.

In order to test Henrich & Gil-White's [2] assertion that prestige functions to identify targets of social learning, we then asked participants to choose a member of their wider community or country from whom they would like to learn a particular skill, or learn to be like. We made the following prediction:

> H2: Individuals rate those from whom they want to learn in the wider community or country as having high prestige, but not high dominance.

Participants then completed the quiz, first individually and then in their groups. During the group stage, participants were told that they had to discuss and agree on every answer, rather than divide the quiz into sections per person, or by voting on each question. After handing in the group answers, each group member then anonymously voted for one other member of their group (excluding themselves) to take part in a bonus quiz on the group's behalf. Nominations were anonymous and private to obtain honest assessments from participants, and avoid social influence in voting decisions that would emerge if voting were public. Before the nominee was revealed, each participant rated each other member of their group for influence, prestige, dominance and likeability. The nominee then completed the bonus round of questions alone.

Following Cheng *et al.*'s finding that prestige and dominance independently predict social influence within groups, we predicted that:

> H3: Individuals who are rated as highly influential in the group task are also rated as highly prestigious or dominant, but not both.

Given that more knowledgeable individuals should score higher on our quiz, and knowledgeable individuals should be prestigious (but not dominant; [2]), we further predicted that:

> H4: Individual score on the quiz predicts prestige but not dominance ratings.

We also tested two predictions stemming from prior work relating to other traits differentially associated with prestige and dominance, but with our acquainted, naturally occurring groups. Given previous findings that prestigious, but not dominant, individuals are liked more [1], we predicted that:

H5: Ratings of high prestige predict likeability, but high dominance ratings do not.

We also predicted that overconfidence would be related to dominance, given previous findings that dominance is related to hubristic rather than authentic pride [7]. We measured overconfidence by asking each participant to estimate their score on the individual section of the quiz. Hence:

H6: Overconfidence predicts ratings of high dominance but not prestige, whereas accurate confidence or underconfidence predicts prestige.

Finally, we made a prediction related to the nomination of one group member to represent the group in the bonus round. Given that participants knew that more knowledgeable group members were more likely to win prize money for their group during the bonus round, and the aforementioned link between knowledge and prestige, we therefore predicted that they should nominate prestigious individuals, hence:

H7: Individuals who are rated as highly prestigious (rather than highly dominant) are nominated to represent the group.

# 2. Methods

All methods and predictions, along with analysis plans, were pre-registered and are available to view on the Open Science Framework at https://osf.io/tasu5/.

## 2.1. Participants

Thirty-two groups of five participants from local, already-established community groups were recruited. These included sports clubs, volunteer groups, businesses, bands and chess clubs. For a full list of participating groups, see electronic supplementary material. Each group had exactly five individuals to control levels of individual influence within each group and allow robust comparison across groups. Groups were recruited between June 2017 and February 2018. Two groups' data were removed. One group (Group number 1) was a pilot test consisting of fellow office members who may have been knowledgeable of our hypotheses. Another group (Group number 11) was not used as an elderly participant left the study halfway through due to concentration problems, leaving the remaining group with only four participants. As we wanted every group to have exactly five participants, we allowed the group to complete the rest of the task, but only used their wider community ratings, and not their within-group ratings. Our participants include 82 females and 54 males (not all participants disclosed their gender), with a mean age of 46.49 (s.d.: 21.51).

## 2.2. Materials

We used the prestige and dominance scales from Cheng *et al*. [1], as well as the measures of 'influence' and 'likeability' used in Cheng *et al*.'s [1] study. All measures are reproduced in our electronic supplementary material.

Our focal task was a general knowledge quiz consisting of 40 alternative choice questions from four categories of 10 questions. Four questions from each category were paired with a picture, the rest were exclusively text-based. The categories were 'Geography', 'Weight estimation', 'Language identification' and 'Art history'. An example question for the weight estimation category is 'How much does a camel weigh?' (a) 48 kg (b) 480 kg. An example question from the language identification category is 'The word 'pisică' means 'cat' in which language? (a) Romanian (b) Hungarian. The four topics were chosen to represent different domains of knowledge that are not necessarily linked to academic ability or education, but may reflect experiences related to the particular domains. For example, an individual may not be particularly educated in mathematics, but may be skilled at estimating weights of various objects due to the nature of their work/hobbies. The 40 questions were based on a freely available database of 15 000 trivia questions, and were adapted to be multiple-choice answers and to fit into the four topics described. These 40 questions were chosen to encompass the hardest and easiest questions from each topic after piloting online. The mean individual score of our 150 participants was 28.625 (3.858). The quiz is available in electronic supplementary material.

## 2.3. Procedure

Participants were given an information sheet and asked to sign their consent before taking part. The experiment was granted ethical approval by the University of Exeter Biosciences ethics committee. Participants were first given stickers with unique ID numbers so that participants never referred to each other by name during the ratings, and ratings were anonymous to each other as well as to the experimenter (C.O.B.). They then completed the following activities:

(1) *Wider community ratings.* Participants were individually asked to name a person either from their local community (e.g. doctor or councillor) or country (e.g. prime minister or celebrity) who they think has high status/influence over their community or country. Participants then rated this person using the prestige and dominance scales. Participants were then asked to name someone either from their local community or country who they would like to 'learn from', or 'learn to be like', and then rated this person using the prestige and dominance scales.

(2) *Initial within-group ratings.* Participants anonymously named another group member (using the ID numbers provided) who they deemed to have 'high status or influence' over the group, e.g. a teacher, tutor, team captain or group administrator. Participants were then asked to name an individual from their group (using the ID numbers) who they would most like to 'learn from' or 'learn to be like' (e.g. the best rower or chess player). Participants were told that if no individual group member immediately comes to mind or fits the criteria then to leave those questions blank.

(3) *Individual quiz.* Participants were then asked to complete the 40 item quiz individually without consulting each other or discussing the questions. After they finished, participants were asked to provide their estimated score on the quiz (i.e. how many questions they answered correctly out of 40) at the bottom of the answer sheet. Participants then handed in their answer sheet marked with their individual ID number.

(4) *Team quiz.* Participants then had 10 min to complete the same 40 question quiz as a group. Participants were told that the highest scoring group will be given a prize of £500. Participants were told that they had to discuss and agree on every answer, rather than divide the quiz into sections per person, or by voting on each question. The experimenter (C.O.B.) told each group that she did not want to affect their decisions or interactions, and so separated herself from the group and wore earphones with loud music playing so as not to be able to hear the group discussion. The experimenter timed the group and let them know when they had 5 min remaining, and 2 min remaining. The experimenter ensured that no participants used their mobile phones throughout.

(5) *Nominations.* After handing in their group answers, participants were asked to anonymously vote for who they would like to be put forward to represent their group on a bonus round of eight new questions, two questions from each section. They were informed that the top scoring group on the bonus round would also be rewarded £500. Participants were instructed not to vote for themselves. Participants were given voting slips with their own ID, so the experimenter could check for this.

(6) *Within-group ratings.* After handing in their voting slips, and before the nominee was revealed, participants anonymously rated each of the other group members on the dominance and prestige scale by using their ID numbers, as well as rating each other group member on how much influence they had on the group decision, and how likeable they were, using the scales used by Cheng *et al.* [1]. Once participants had finished the rating sheets, the nominated individual was revealed and given the bonus quiz to complete individually. Participants were then debriefed. The winning groups were announced and awarded the prize money in March 2018 after all groups had been tested.

## 2.4. Analyses

We used a model comparison approach with a variety of multi-level models using the *Rethinking* package in R [9]. Full analysis scripts and data are available at www.github.com/lottybrand22/GH_Kernow, and were pre-registered at https://osf.io/tasu5/.

   Model comparisons were made by comparing the models' widely applicable information criteria (WAIC) values. Models were said to be a better fit to the data if their WAIC value held the most weight out of all models tested. Model parameters were said to have an effect on the model outcome if their 89% credible interval did not cross zero. The 89% intervals are the default credible interval setting for the Rethinking package [9]. The 95% intervals would not alter the interpretation of our results. We included model parameters based on pre-registered *a priori* hypotheses. All predictors were centred and scaled. See tables 1–3 for the model specifications including parameters for all models.

**Table 1.** Model comparison for the prestige models, with prestige ratings by other group members as the predicted outcome in the model.

| model | parameters | WAIC | weight | s.e. |
|---|---|---|---|---|
| full | score + influence + likeability + confidence + initially influential + learning model + age + sex + nominated + 1\|scale item + 1\|RaterID + 1\|RatedID + 1\|Group | 11034.1 | 0.65 | 124.13 |
| *a priori* | score + influence + likeability + 1\|scale item + 1\|RaterID + 1\|RatedID + 1\|Group | 11035.3 | 0.35 | 123.65 |
| null | 1\|scale item + 1\|RaterID + 1\|RatedID + 1\|Group | 11053.3 | 0.00 | 123.63 |
| exploratory | initially influential + learning model + 1\|scale item + 1\|RaterID + 1\|RatedID + 1\|Group | 11054.7 | 0.00 | 123.92 |

**Table 2.** Model comparison of the dominance models, with dominance ratings by other group members as the predicted outcome in the model.

| model | parameters | WAIC | weight | s.e. |
|---|---|---|---|---|
| full | confidence + influential + score + likability + nominated + initially influential + learning model + sex + age + 1\|scale item + 1\|RaterID + 1\|RatedID + 1\|Group | 10373.9 | 0.59 | 114.43 |
| *a priori* | confidence + influential + 1\|scale item + 1\|RaterID + 1\|RatedID + 1\|Group | 10375.4 | 0.28 | 114.52 |
| null | 1\|scale item + 1\|RaterID + 1\|RatedID + 1\|Group | 10377.0 | 0.12 | 114.77 |
| exploratory | initially influential + learning model + 1\|scale item + 1\|RaterID + 1\|RatedID + 1\|Group | 10383.4 | 0.01 | 114.75 |

**Table 3.** Model comparison for the nominations models, with likelihood of being nominated to represent the group in the bonus round as the predicted outcome.

| model | parameters | WAIC | weight | s.e. |
|---|---|---|---|---|
| full | intercept + score + confidence + prestige + dominance + influence + likeability + initially influential + learning model + 1\|Group | 108.6 | 1 | 13.43 |
| score model (exploratory) | intercept + score + confidence + 1\|Group | 122.8 | 0 | 13.22 |
| influence model | intercept + influence + 1\|Group | 130.3 | 0 | 11.00 |
| dominance model | intercept + dominance + 1\|Group | 138.6 | 0 | 12.64 |
| previous relationships model | intercept + initially influential + learning model + 1\|Group | 138.9 | 0 | 12.59 |
| null model | intercept + 1\|Group | 140.6 | 0 | 12.37 |
| prestige model | intercept + prestige + 1\|Group | 141.0 | 0 | 12.68 |
| likeability model | intercept + likeability + 1\|Group | 142.7 | 0 | 12.78 |

Priors were chosen to be weakly regularizing, in order to control for both under- and overfitting the model to the data [9]. The robustness of the results was checked by trying a variety of priors, as well as a variety of chains and iterations. Trace plots and effective sample sizes were used to check for appropriate model convergence throughout.

When analysing the prestige and dominance Likert scale ratings, we used ordinal categorical multi-level models, with varying intercepts for who the rater was (rater ID), who was being rated (rated ID), the group and Likert scale item. This allowed us to use each Likert scale item as the unit of analysis, rather than average over several items, in accordance with recent recommendations on how to treat Likert scale data [10,11].

When analysing who was nominated for the bonus round, we used a binomial multi-level model with group as a varying intercept. When prestige and dominance ratings were used as predictor variables in the nomination model, the Likert scale ratings for each person were averaged and a proportion of the total scale was calculated for each participant. Thus, if a person was given 7/7 for every item by every rater, they would have a total proportion of 1, and if a person was given 1/7 for every item by every rater, they would have a total proportion of 0.

Overconfidence was calculated as the participants' estimated score subtracted from their actual score. Thus, any positive value reflects 'overconfidence', and a negative value would reflect 'underconfidence'. In addition, we coded the variable as binary to reflect the prediction that 'accurate confidence or underconfidence' would be related to prestige. As 'accurate' confidence is only represented by a value of zero (no difference between estimated and achieved score), we coded overconfidence as anything over zero, and everything else as 'accurate or underconfidence'. This binary measure did not give different results to our continuous measure, so we report our continuous, pre-registered measure in our Results section.

Participants were coded as 'initially influential' (1 rather than zero) if they were named by at least three group members in the initial within-group ratings as someone who is particularly influential to the group, such as a team captain or group administrator. Participants were coded as 'learning models' (1 rather than zero) if they were named by at least three group members in the initial within-group ratings as someone from whom they would like to learn from, e.g. learn a skill from or learn to be like. When sex was a predictor in any model, males were coded as zero and females as 1, thus any effect of sex is the effect of being female compared to male.

# 3. Results

## 3.1. H1 and H2: wider community ratings of prestige and dominance

Hypotheses H1 and H2 concerned the participants' prestige and dominance ratings of community figures. H1 was partially supported, in that influential members of the community, such as politicians, activists or celebrities, were rated as more prestigious than dominant (mean (s.d.) prestige = 4.88 (1.33); dominance = 3.89 (1.45)). Contrary to H1, and to Cheng *et al*. [1], prestige and dominance ratings for influential community figures were not statistically independent, but instead dominance ratings were negatively predicted by prestige ratings (mean coefficient estimate: −3.22, 89% confidence interval: [−3.86, −2.61]). That is, figures who were rated as highly prestigious were rated as low in dominance, and vice versa. For a full list of influential figures who were named, see electronic supplementary material. Cronbach's $\alpha$ for both the prestige and the dominance scale was high (prestige: $\alpha = 0.89$; dominance: $\alpha = 0.86$), showing a high level of internal consistency in the scales.

In support of H2, members of the community from whom participants would like to learn were rated as highly prestigious, but not highly dominant (mean (s.d.) prestige = 5.77 (0.70); dominance = 2.98 (1.19)). Prestige and dominance ratings for these community learning models were statistically independent, given that prestige ratings did not predict dominance ratings (coefficient estimate: −0.84, 89% CI: [−1.88, 0.14]). For a full list of community members from whom participants would like to learn, and the skills they would like to learn, see electronic supplementary material.

## 3.2. H3−H6: within-group prestige and dominance ratings

To test hypotheses H3−H6, we ran two sets of model comparisons, one with prestige ratings from other group members as the outcome (figure 1 and table 1), and the other with dominance ratings from other group members as the outcome (figure 3 and table 2).

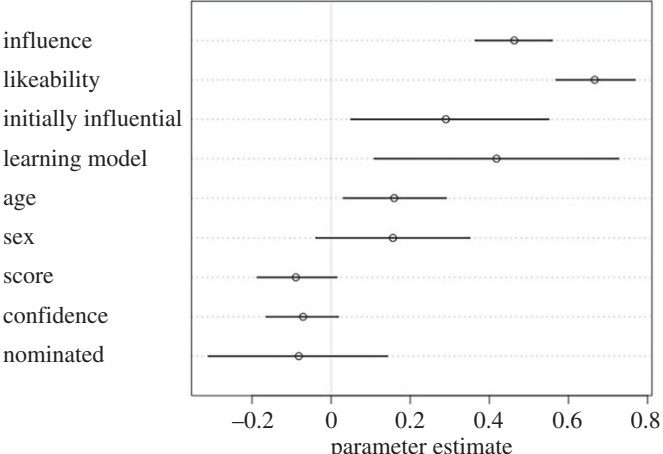

**Figure 1.** Parameter estimates for the full prestige model, with prestige ratings by other group members as the outcome. Estimates that cross zero suggest that parameter did not have a strong effect on prestige ratings.

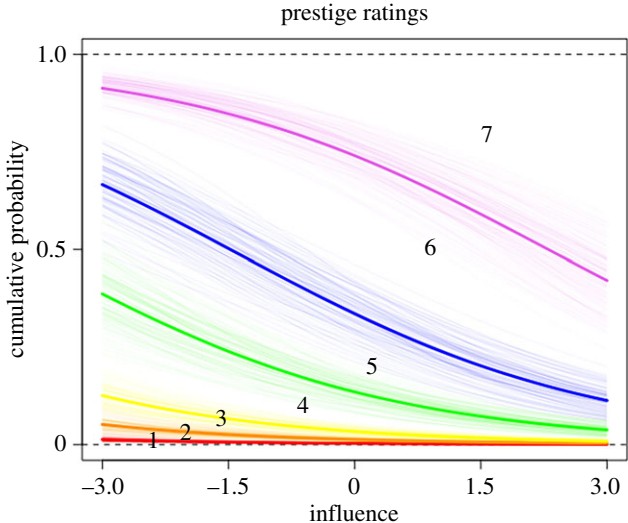

**Figure 2.** Posterior predictions of the ordered categorical prestige model showing how the distribution of each Likert scale response varies with influence ratings. The lines indicate boundaries between response values, numbered 1–7, with thick lines indicating the mean prediction for that boundary. For example, the space between the purple and blue lines marked with a '6' indicates the relative number of participants who gave a rating of 6. The figure shows that if a participant is rated as highly influential, they are more likely to also be given a high prestige rating (e.g. 7) compared to less influential participants.

In accordance with H3, participants were rated as more prestigious if they were also rated as highly influential during the group task (mean coefficient estimate: 0.46, 89% CI: [0.36, 0.55], figure 2). Contrary to H4, however, individual quiz score did not predict high prestige ratings (score: −0.09, CI: [−0.19, 0.01]). In accordance with H5, participants were more likely to be rated as highly prestigious if they were also rated as highly likeable (0.67, CI: [0.57, 0.77]). Contrary to H6, confidence on the quiz was unrelated to prestige ratings (−0.07, CI: [−0.16, 0.02]) (figure 1).

In addition to our *a priori*, hypothesis-based prestige model, results from our full model suggested that participants were more likely to be rated as highly prestigious if they were initially named as an influential group member, such as a team captain or group administrator (0.29, 89% CI: [0.05, 0.55]). Participants were also rated as more prestigious if they were initially named as someone from whom others would like to learn (0.41, CI: [0.11, 0.73]). Whether the participant was nominated for the bonus round did not predict prestige ratings, neither did the sex of the participant (nominated: −0.08, CI: [−0.31, 0.14], sex: 0.16, CI: [−0.04, 0.35]). However, age did predict prestige ratings, with older participants rated as more prestigious (0.16, CI: [0.03, 0.29]) (figure 1).

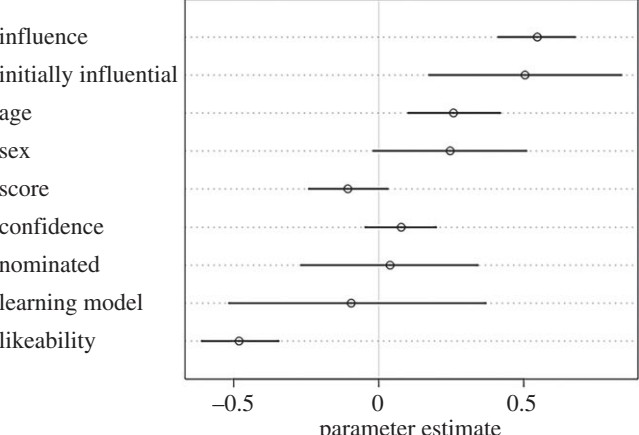

**Figure 3.** Parameter estimates of the full dominance model, with dominance ratings by other group members as the outcome. Estimates that cross zero suggest that parameter did not have a strong effect on dominance ratings.

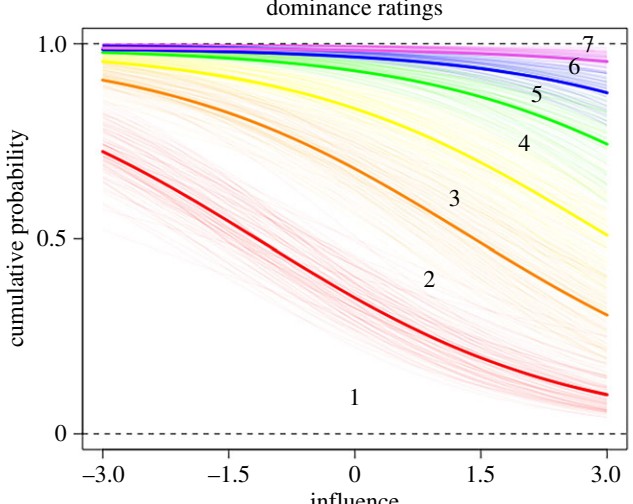

**Figure 4.** Posterior predictions of the ordered categorical dominance model showing how the distribution of each Likert scale response varies with influence ratings. The lines indicate boundaries between response values, numbered 1−7, with thick lines indicating the mean prediction for that boundary. For example, the space between the orange and red lines marked with a '2' indicates the relative number of participants who gave a rating of 2. The figure shows that if a participant has low influence, they are more likely to be given a low dominance rating (e.g. 1) compared to highly influential participants.

The best fitting model was the full model, suggesting that other parameters were important aside from those in the *a priori*, hypothesis-based model. An exploratory model was included in addition to the pre-registered models, to see if initial ratings produced a better model fit than the prestige model; however, this was not the case (table 1).

Turning to the dominance models (figure 3, table 2), in accordance with H3, participants were more likely to be rated as highly dominant if they were also rated as highly influential during the group task (0.55, CI: [0.41, 0.68], figure 4). Further analysis showed that dominance ratings were not predicted by prestige ratings, and thus, dominance and prestige ratings were statistically independent from each other (0.71, CI: [−0.27, 1.69]), in line with H3. In accordance with H4, score on the quiz did not predict dominance ratings (−0.11, CI: [−0.24, 0.03]). In contrast with H5, likeability negatively predicted dominance in that lower likeability ratings predicted higher dominance ratings (−0.48, CI: [−0.61, −0.35]). Contrary to H6, however, overconfidence on the quiz did not predict higher dominance ratings (0.08, CI: [−0.05, 0.20]) (figure 4).

Aside from our *a priori*, hypothesis-based model, results from our full model suggested that participants were more likely to be rated as highly dominant if they were initially rated as an

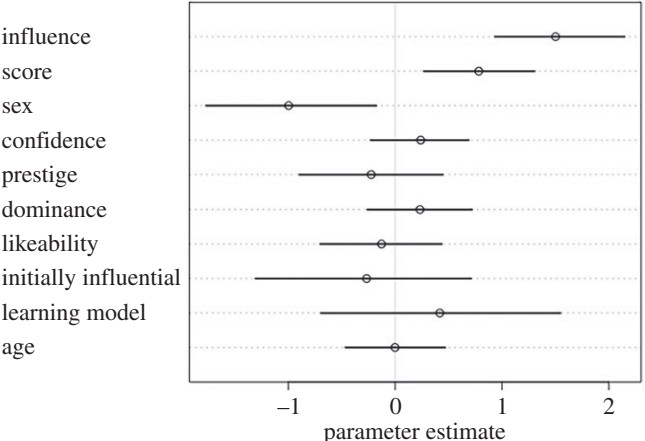

**Figure 5.** Parameter estimates for the full nominations model, with likelihood of being nominated to represent the group in the bonus round as the outcome. Estimates that cross zero suggest that parameter did not have a strong effect on the likelihood of being nominated.

influential group member (0.50, CI: [0.17, 0.84]), and if they were older (0.26, CI: [0.10, 0.42]). Whether the participant was initially named as someone from whom others would like to learn did not predict high dominance ratings (−0.09, CI: [−0.52, 0.37]), nor did sex (0.25, CI: [−0.02, 0.51]), or whether the participant was nominated for the bonus round (0.04, CI: [−0.27, 0.34]) (figure 4).

The best fitting model was the full model, suggesting that other parameters were important aside from those in the *a priori*, dominance-based model. An exploratory model was compared in addition to the pre-registered models, to see if initial ratings produced a better model fit than the dominance model; however, this was not the case (table 2).

## 3.3. H7: nominations

Hypothesis H7 was tested using nomination for the bonus quiz round as the outcome measure. While dominance failed to predict nominations for the bonus round as predicted, nor did prestige, which was expected to predict nominations (prestige: −0.22, CI: [−0.90, 0.45], dominance: 0.23, CI: [−0.26, 0.72]) (figure 5).

Aside from our *a priori* hypotheses, the full model suggested that participants were more likely to be nominated for the bonus round if they scored highly on the individual quiz (0.78, CI: [0.27, 1.30]). Neither likeability, overconfidence on the quiz, nor age were related to nominations for the bonus round (likeability: −0.13, CI: [−0.70, 0.44], confidence: 0.24, CI: [−0.23, 0.69], age: 0.00, CI: [−0.46, 0.47]). Whether an individual held an influential position in the group, or was someone from whom others would like to learn, also had no relationship to nominations for the bonus round (initially influential: −0.27, CI: [−1.30, 0.71], learning model: 0.42, CI: [−0.69, 1.55]). However, participants were more likely to be nominated if they were rated as highly influential during the group quiz discussion (1.50, CI: [0.94, 2.15]), and less likely if they were female (−1.00, CI: [−1.79, −0.18]) (figure 5).

The best fitting model was the full model, suggesting that other parameters were important aside from those in our *a priori* model. An exploratory model was also tested to see if score and confidence on the quiz provided a better fit than the other *a priori* models; however, this was not the case (table 3).

## 3.4. Summary of hypotheses

In summary, H1 was partially supported, in that prestige and dominance ratings for influential wider community figures were not statistically independent, but they were negatively related, i.e. high prestige was associated with low dominance and vice versa. H2 was supported in that wider community figures from whom participants would like to learn have high prestige and low dominance, and prestige and dominance were unrelated. H3 was supported: influential group members were rated as highly prestigious or dominant, and these were statistically independent. H4 was partially supported: while dominance was unrelated to quiz score as predicted, prestige was also unrelated, contrary to our predictions. H5 was partially supported: likeability was positively related to

prestige as predicted, but dominance was negatively related to likeability, contrary to our prediction of no association. H6 was unsupported: confidence was unrelated to both prestige and dominance. Finally, H7 was partially supported: while dominance did not predict nominations for the bonus quiz, as expected, neither did prestige, contrary to our predictions.

# 4. Discussion

We ran an experiment testing multiple aspects of theories of prestige and dominance with 30 naturally occurring groups in Cornwall. Participants completed a quiz individually and then as part of a team to win prize money. Participants anonymously voted for one team member to complete a bonus quiz on behalf of the group, providing a payoff-related, 'behavioural' measure of influence beyond self-reported ratings. Participants also rated members of their wider community using scales of prestige and dominance, and anonymously rated the other members of their team using the same scales. Our methodology is novel in (i) using a demographically wide sample of participants within existing groups rather than university students brought together for the purpose of an experiment, and (ii) using a task explicitly tied to knowledge, allowing us to test theorized links between prestige and knowledge (and the predicted absence of a link between dominance and knowledge).

We found that individuals' ratings of influence during the group quiz discussion were related to ratings of prestige and dominance, as found in previous studies [1]. Within-group ratings of prestige and dominance were also statistically independent from each other, as was found in previous studies [1]. However, contrary to our predictions, ratings of prestige and dominance did not predict who was voted for as team representative for the bonus quiz (our 'behavioural' measure of influence). Instead, participants voted for those who scored highly on the individual quiz, even though this information was unavailable to them. This suggests individuals were accurately able to assess others' knowledge during the team discussion, and reliably vote for those who had the most knowledge, yet this knowledge did not translate to higher prestige ratings.

Although prestige and dominance ratings were unrelated to team nominations, we did find an unexpected relationship between initial ratings of within-group influence, and prestige and dominance ratings. That is, individuals who were initially named as having an influential role in the group such as team captain or group administrator, had higher prestige and dominance ratings than their team-mates.

Taken together, these findings suggest that prestige and dominance may be more domain-specific, or more fixed, than we had anticipated. As teams were already established, rather than being groups of unacquainted strangers, prestige and dominance ratings were better predicted by the individual's role in the group, rather than their performance on the quiz. Consequently, when voting for group representatives for a bonus quiz, group members did not vote for those with influential roles or high prestige and dominance, but voted for those with the highest quiz score. This suggests that prestige and dominance hierarchies develop over time within a group, but that these perceptions of dominance and prestige are not easily altered on a short-term basis, such as within the duration of our experiment. Alternatively, prestige could be highly domain-specific, and thus, the prestige attained by showing expertise or knowledge in the activity practised by the group (e.g. knitting or playing chess) did not transfer to the general knowledge required to answer our quiz questions. Previous findings that prestige predicts performance on tasks (e.g. [1]) may therefore partially be a product of bringing unacquainted groups of strangers together with no prior relationships. In the absence of such prior relationships, discussions during the experiment are the only basis for prestige and dominance perceptions to develop. This result also highlights the need for researchers to control for, and be cautious of, any existing relationships within participant samples, as these may interfere or override any short-term manipulations within the experimental set-up. Indeed, a recent study found that although prestige and dominance are both related to social status in short-term interactions, prestige appears to be more important over long-term interactions [12]. Future studies might also further explore the domain-specificity of prestige hierarchies, for example, by varying and examining the match between a group's specific activity (e.g. chess, kayaking) and the experimental task. Our quiz was designed to tap general knowledge and be unrelated to group activities; however, if group activity matched the task (e.g. a group of pub quizzers), then we might expect prestige ratings to more accurately predict nominations.

Interestingly, both members of the wider community, and members of the groups, who participants named as someone from whom they would like to learn a skill, or learn to be like, were rated as highly

prestigious, but not highly dominant, as we predicted. This finding supports the theory that prestige evolved in the context of social learning, in that individuals who made particularly good learning models also attained prestige from other members of their group [2]. Likewise, our findings support the theory that dominant individuals do not make good learning models, as their status is attained via threat or fear rather than through skill or knowledge. Another distinguishing feature of individuals with high prestige but not high dominance ratings was likeability, in that individuals who were rated as highly likeable had higher prestige ratings on average, but lower dominance ratings. Indeed, high dominance ratings predicted low likeability ratings. This also supports previous studies that found highly prestigious individuals are liked, whereas highly dominant individuals are not liked [1].

Contrary to our predictions, overconfidence on the quiz did not predict dominance ratings. A previous study found that prestige was related to authentic pride, while dominance was related to hubristic pride [7]. However, neither confidence, nor overconfidence in one's ability, predicted prestige or dominance ratings in our task. Further work is needed to understand whether certain personality factors contribute to an individual's likelihood of being perceived as prestigious or dominant, and whether confidence is related to an individual's likelihood of gaining prestige or dominance. It is worth noting that in our study, prestige and dominance were related to positions of influence in the group that were attained before our task, and that performance on our task was not related to prestige and dominance ratings. Thus, the lack of relationship between confidence on the quiz and prestige and dominance ratings in our case may be explained by the lack of relationship between quiz performance and dominance/prestige ratings.

We did not have any specific predictions about sex or age in our study; however, our full models included effects of sex and age and were better-fitting models than our hypothesis-based models. Age was related to prestige, in that older individuals were rated as more prestigious than younger individuals. This supports previous findings showing that age is positively related to prestige [13,14], and supports the theory that older members of groups possess valuable skills and knowledge that earn them prestige compared with younger members (although other studies have failed to find support for this idea [14]). Age was also related to dominance ratings, with older individuals rated as more dominant than younger individuals. Interestingly, women were less likely to be nominated for the bonus round than men. This finding is particularly hard to interpret, given that women's individual quiz scores and likeability ratings were no different to men's; however, it may in part be due to women's lower average confidence than men's, which is a widely reported sex difference in a variety of domains [15–17]. However, it is important to note that this result was not part of our original predictions, thus these are only speculative, post hoc explanations. Further study and experimental evidence is required to interpret these results with more certainty.

Worthy of note is that our study included a knowledge-based task, a quiz, rather than the skill-based tasks used in many previous studies of social learning such as flint-knapping, knot-tying, basket-weaving or spaghetti-tower building [18–21]. Although Henrich & Gil-White did not distinguish between knowledge and skill in their original discussion of prestige [2], we feel it is an important distinction to make and a necessary avenue of future research. One reason for the importance of this distinction is the potential difference in observation and learning opportunities for manual skills versus abstract knowledge. The majority of current evidence of social learning and cultural evolution in humans comes from either ethnographic observations, or experimental tasks, in which a manual skill is learnt via observation such as imitation or emulation. However, it can be argued that the majority of learning opportunities are not via direct imitation of a manual skill, but by acquiring knowledge, often via language and explicit teaching, particularly in contemporary, post-industrialized societies with formal education systems and knowledge-based economies. To what extent language interferes with, or enhances, the social learning of knowledge (and indeed skills) needs to be addressed. It is possible that success and prestige biases are crucial when socially learning skills, but the transmission of knowledge may be governed by alternative social learning strategies that are not currently considered. Future research, using social learning experiments with language-mediated knowledge rather than skill as the focus, may help to address this distinction further.

In conclusion, we have found evidence that prestige and dominance hierarchies do exist in naturally occurring groups in a diverse, adult population. Although prestige ratings were not related to knowledge (quiz score), they were related to whether the individual held an influential position in the group already (such as team captain or group administrator). Furthermore, nominations for the bonus quiz were not predicted by prestige or dominance ratings, but were instead best predicted by the individual's score on the quiz. We interpret this as potential evidence of the domain-specificity of prestige and dominance hierarchies, in that individuals who had attained their prestige or dominance through the

group's regular activity were not nominated to represent the group on an unrelated task (the quiz). We encourage further work exploring the applicability of prestige and dominance measures in demographically diverse samples, as well as the theorized but under-studied link between prestige and knowledge. Finally, we recommend further investigation of the domain-specificity and generality of prestige and dominance.

Ethics. The experiment was granted ethical approval by the University of Exeter Biosciences ethics committee, application number 2017/1979.

Data accessibility. Our data and analysis scripts are available at www.github.com/lottybrand22/GH_Kernow. Pre-registered on the Open Science Framework at https://osf.io/tasu5/.

Authors' contributions. C.O.B. and A.M. designed the project. C.O.B. recruited participants, ran the experiments, collected and recorded the data and analysed the data. A.M. helped with plotting the data. C.O.B. and A.M. wrote and edited the manuscript. Both authors gave final approval for publication.

Competing interests. The authors declare no competing interests.

Funding. This research was supported by The Leverhulme Trust (grant no. RPG-2016-122 awarded to A.M.)

Acknowledgements. Thank you to Joey Cheng, Stefan Gehrig, Matt Gobel and Ángel Jiménez for useful comments on previous drafts. We thank Amanda Lucas and Devi Whittle for sharing their contacts of Cornwall-based community groups for participation in our experiment. Finally, we thank the community groups who gave up their time to participate in our study, and the Royal Cornwall Polytechnic Society (The Poly), Falmouth for hosting a public event to disseminate our findings.

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
