## [Reviewer comments · Royal Society Open Science]

Review History

RSOS-181621.R0 (Original submission)

Review form: Reviewer 1 (Robin Dunbar)

Is the manuscript scientifically sound in its present form?

Yes

Are the interpretations and conclusions justified by the results?

Yes

Is the language acceptable?

Yes

Is it clear how to access all supporting data?

Yes

Do you have any ethical concerns with this paper?

No

Have you any concerns about statistical analyses in this paper?

No

Recommendation?

Accept with minor revision (please list in comments)

Comments to the Author(s)

This is neat little study and a well written paper. I have only a couple of comments that I think could be addressed.

(1) l. 647: it is surely doubtful that the only thing people learn in ethnographic societies is manual skills! Knowledge is equally important, not to mention acquiring group-specific cultural knowledge like folktales.

(2) much is made in the Introduction about the fact that this study will look at natural groups rather than groups of strangers (as in the Chen study -- and indeed, most experimental studies in the social sciences). Yet hardly anything is made of this in the Discussion. This is surely the most interesting aspect of the whole study? Almost everything in this area is done with strangers on the apparent assumption that our lives consist in buying secondhand cars from untrustworthy strangers. This might sometimes be the case in economics, but is simply isnt that common in real social life, most of which involves people we know personally and only very occasionally involves strangers. So perhaps a bit more discussion of the differences between the Cheng study and this one in the light of this would be in order? If there are significant differences (as there were in Henrich's tribal economic games studies), this has very important implications for the way we do any experiments in the social sciences.

Review form: Reviewer 2

Is the manuscript scientifically sound in its present form?

Yes

Are the interpretations and conclusions justified by the results?

Yes

Is the language acceptable?

Yes

Is it clear how to access all supporting data?

No

Do you have any ethical concerns with this paper?

Yes

Have you any concerns about statistical analyses in this paper?

Yes

Recommendation?

Major revision is needed (please make suggestions in comments)

Comments to the Author(s)

This manuscript examines the emergence of prestige and dominance perceptions among small group members in “naturally occurring” human groups. Thirty groups of five members formed on the basis of specific interests (e.g., chess, knitting) were tested individually on general knowledge quizzes, had an opportunity to discuss and submit quiz answers as a team, and then nominated a representative to answer additional quiz questions. All group members also rated each group member, as well as members of the broader community, on dimensions of prestige, dominance, and influence. The authors find that higher ratings on prestige and dominance were associated with preexisting group leaders, but these ratings did not predict who would be nominated as group representative. Rather, the group representative nomination was associated with the highest individual quiz score, although the individual scores were not known.

I read this manuscript with interest, and I appreciated the efforts taken to obtain a diverse community sample of groups with varying sets of interests. I also appreciated how the authors reported both supported and unsupported hypotheses in an unbiased way. I had some questions and comments that I think would help to clarify some aspects of the paper.

Hypotheses:

It would be helpful if the authors could explain some more about the theorizing regarding the prestige and dominance constructs. I understand that in previous work (Cheng et al.) prestige and dominance were shown to be independent such that “influential group members are either prestigious or dominant, but not both” (p. 12). I also understand that prestige and dominance are separable dimensions, but it is not as clear to me why they should be independent (H1, p. 10). For instance, the authors show in the current data that prestige and dominance ratings are negatively associated. In addition, it seems plausible that there are examples of figures who exhibit both traits, people who have a high level of expertise and a dominating personality, as well as, conversely, people who have low expertise and a more subordinate personality. I am also a bit confused about some of the examples given for each of these dimensions – why is a teacher an example of a dominant/high status person as opposed to a person from whom to learn (p. 18)?

Almost all the predictions are about prestige, and it seems the only prediction regarding dominance is about overconfidence. This left me wondering what the utility of examining dominance is. Theoretically, why do group members tolerate dominant members? Do dominant members serve some purpose, such as being perceived as providing more existential safety from outgroup threats compared to prestigious members (who perhaps serve more epistemic needs)? Or are dominant members tolerated simply because they are feared and might otherwise serve an existential threat?

Relatedly, I wondered whether there could be a public/private dimension that affects the nominations. That is, would nominations that are made publicly (i.e., in front of the group) be driven by dominance ratings?

Theory/Design:

As mentioned earlier, I like that the authors examined existing interest-based groups for greater ecological validity than arbitrarily assigned student groups. However, the advantages of examining the existing groups seem somewhat lost with the current design. That is, it's not clear to me what we learn about naturally occurring groups if they are essentially treated as arbitrary groups. I would have thought that it is potentially more useful to understand the emergence and existence of prestige and dominance perceptions within the context of each group's interest. For example, for a group formed on the basis of interest in chess, it seems that assessments of prestige, dominance, influence, and learning opportunities would be best assessed on the topic of chess. As the authors acknowledge, it appears that participants' judgments were based on

preexisting attitudes formed in the context of the group's organizing interest and prior interactions, and it seems such attitudes are stronger than the immediate study context of general knowledge quizzes. Presumably group members don't know how generally knowledgeable each other are until they have the group discussion, which makes them functionally similar to the arbitrary college groups in prior work. This point is touched upon in the discussion, but it seems a more important consideration for the paper's theorizing to me.

As the authors note, the broader hypotheses about prestige are not supported in this study in that higher prestige ratings in the group do not translate to higher perceptions of potential learning, likely because prestige ratings seem to be based on preexisting attitudes, whereas the social learning is specific to the general knowledge quiz. It would be helpful if the authors could elaborate on the potential short-term vs. long-term or context-specific relationship between prestige and social learning.

Analyses:

I have a few small comments/suggestions that I think make the analyses easier to understand for readers who may not have expertise with multilevel models (for example, me):

I think the analyses section could be more helpful if it were integrated into the results. As it is now, the analysis section reads as a list of separate decisions that aren't so easy to follow because it's not always obvious what they are for (e.g., on p. 23 it is stated that "models were said to be a better fit to the data if they had WAIC value..." but I think the model has not yet been specified. What are the variables being accounted for? In fact, I don't think I saw any section in which the model was clearly specified. For instance, was dominance regressed on prestige? What are other covariates being used?).

In general, I think it would be helpful to get more explanation about the analysis decisions. For example, could the authors describe how the decision was made to have an 89% credible/confidence interval? I understand that the typical intervals of 90 or 95% are somewhat arbitrary cutoffs but it seems 89% would need to be justified somehow, especially since it seems the ranges generally aren't close to crossing zero (e.g., p. 25).

More minor suggestions:

I thought more information from the supplementary materials would be more useful in the main text. For instance, while reading the main text, I was wondering what the different group types were – perhaps those could be listed in a main text table. This seems important information given the emphasis on the use of novel group types compared to previous studies. Additionally, I thought it would be helpful to include (in the main text) a few examples of the types of items used to measure prestige and dominance, as well as the scale ranges. I was confused by the description of prestige vs. dominance ratings that were approximately 1 point apart but were described as demonstrating perceptions as "highly prestigious but not highly dominant" (p. 25).

Decision letter (RSOS-181621.R0)

09-Jan-2019

Dear Dr Brand,

The editors assigned to your paper ("Prestige and dominance based hierarchies exist in naturally

occurring human groups, but are unrelated to task-specific knowledge") have now received comments from reviewers. We would like you to revise your paper in accordance with the referee and Associate Editor suggestions which can be found below (not including confidential reports to the Editor). Please note this decision does not guarantee eventual acceptance.

Please submit a copy of your revised paper before 01-Feb-2019. Please note that the revision deadline will expire at 00.00am on this date. If we do not hear from you within this time then it will be assumed that the paper has been withdrawn. In exceptional circumstances, extensions may be possible if agreed with the Editorial Office in advance. We do not allow multiple rounds of revision so we urge you to make every effort to fully address all of the comments at this stage. If deemed necessary by the Editors, your manuscript will be sent back to one or more of the original reviewers for assessment. If the original reviewers are not available, we may invite new reviewers.

- Data accessibility

If you wish to submit your supporting data or code to Dryad (<http://datadryad.org/>), or modify your current submission to dryad, please use the following link:
<http://datadryad.org/submit?journalID=RSOS&manu=RSOS-181621>

- Competing interests

- Authors' contributions

- Acknowledgements

- Funding statement

on behalf of Professor Antonia Hamilton (Subject Editor)
openscience@royalsociety.org

Editor's comments:

We are awarding a major revision decision to your paper as the reviewers have identified a number of matters that must be addressed prior to the paper being considered further for publication. The reviewers and Editors also note that the anonymisation of study participants is incomplete in the electronic supplements -- please look again at this and ensure A) that you check this with your ethical review board and B) respond to this concern (and how you have tackled it) in the response to referees in your revision. If you need a week or two longer to achieve this owing to the time of year, please let the editorial office know.

Comments to Author:

Reviewers' Comments to Author:

Reviewer: 1

Comments to the Author(s)

This is neat little study and a well written paper. I have only a couple of comments that I think could be addressed.

(1) l. 647: it is surely doubtful that the only thing people learn in ethnographic societies is manual skills! Knowledge is equally important, not to mention acquiring group-specific cultural knowledge like folktales.

(2) much is made in the Introduction about the fact that this study will look at natural groups rather than groups of strangers (as in the Chen study -- and indeed, most experimental studies in the social sciences). Yet hardly anything is made of this in the Discussion. This is surely the most interesting aspect of the whole study? Almost everything in this area is done with strangers on the apparent assumption that our lives consist in buying secondhand cars from untrustworthy strangers. This might sometimes be the case in economics, but is simply isn't that common in real social life, most of which involves people we know personally and only very occasionally involves strangers. So perhaps a bit more discussion of the differences between the Cheng study and this one in the light of this would be in order? If there are significant differences (as there were in Henrich's tribal economic games studies), this has very important implications for the way we do any experiments in the social sciences.

Reviewer: 2

Comments to the Author(s)

This manuscript examines the emergence of prestige and dominance perceptions among small group members in "naturally occurring" human groups. Thirty groups of five members formed on the basis of specific interests (e.g., chess, knitting) were tested individually on general knowledge quizzes, had an opportunity to discuss and submit quiz answers as a team, and then nominated a representative to answer additional quiz questions. All group members also rated each group member, as well as members of the broader community, on dimensions of prestige, dominance, and influence. The authors find that higher ratings on prestige and dominance were associated with preexisting group leaders, but these ratings did not predict who would be nominated as group representative. Rather, the group representative nomination was associated with the highest individual quiz score, although the individual scores were not known.

I read this manuscript with interest, and I appreciated the efforts taken to obtain a diverse community sample of groups with varying sets of interests. I also appreciated how the authors reported both supported and unsupported hypotheses in an unbiased way. I had some questions and comments that I think would help to clarify some aspects of the paper.

Hypotheses:

It would be helpful if the authors could explain some more about the theorizing regarding the prestige and dominance constructs. I understand that in previous work (Cheng et al.) prestige and dominance were shown to be independent such that "influential group members are either prestigious or dominant, but not both" (p. 12). I also understand that prestige and dominance are separable dimensions, but it is not as clear to me why they should be independent (H1, p. 10). For instance, the authors show in the current data that prestige and dominance ratings are negatively associated. In addition, it seems plausible that there are examples of figures who exhibit both traits, people who have a high level of expertise and a dominating personality, as well as, conversely, people who have low expertise and a more subordinate personality. I am also a bit confused about some of the examples given for each of these dimensions -- why is a teacher an example of a dominant/high status person as opposed to a person from whom to learn (p. 18)?

Almost all the predictions are about prestige, and it seems the only prediction regarding dominance is about overconfidence. This left me wondering what the utility of examining dominance is. Theoretically, why do group members tolerate dominant members? Do dominant

members serve some purpose, such as being perceived as providing more existential safety from outgroup threats compared to prestigious members (who perhaps serve more epistemic needs)? Or are dominant members tolerated simply because they are feared and might otherwise serve an existential threat?

Relatedly, I wondered whether there could be a public/private dimension that affects the nominations. That is, would nominations that are made publicly (i.e., in front of the group) be driven by dominance ratings?

Theory/Design:

As mentioned earlier, I like that the authors examined existing interest-based groups for greater ecologically validity than arbitrarily assigned student groups. However, the advantages of examining the existing groups seem somewhat lost with the current design. That is, it's not clear to me what we learn about naturally occurring groups if they are essentially treated as arbitrary groups. I would have thought that it is potentially more useful to understand the emergence and existence of prestige and dominance perceptions within the context of each group's interest. For example, for a group formed on the basis of interest in chess, it seems that assessments of prestige, dominance, influence, and learning opportunities would be best assessed on the topic of chess. As the authors acknowledge, it appears that participants' judgments were based on preexisting attitudes formed in the context of the group's organizing interest and prior interactions, and it seems such attitudes are stronger than the immediate study context of general knowledge quizzes. Presumably group members don't know how generally knowledgeable each other are until they have the group discussion, which makes them functionally similar to the arbitrary college groups in prior work. This point is touched upon in the discussion, but it seems a more important consideration for the paper's theorizing to me.

As the authors note, the broader hypotheses about prestige are not supported in this study in that higher prestige ratings in the group do not translate to higher perceptions of potential learning, likely because prestige ratings seem to be based on preexisting attitudes, whereas the social learning is specific to the general knowledge quiz. It would be helpful if the authors could elaborate on the potential short-term vs. long-term or context-specific relationship between prestige and social learning.

Analyses:

I have a few small comments/suggestions that I think make the analyses easier to understand for readers who may not have expertise with multilevel models (for example, me):

I think the analyses section could be more helpful if it were integrated into the results. As it is now, the analysis section reads as a list of separate decisions that aren't so easy to follow because it's not always obvious what they are for (e.g., on p. 23 it is stated that "models were said to be a better fit to the data if they had WAIC value..." but I think the model has not yet been specified. What are the variables being accounted for? In fact, I don't think I saw any section in which the model was clearly specified. For instance, was dominance regressed on prestige? What are other covariates being used?).

In general, I think it would be helpful to get more explanation about the analysis decisions. For example, could the authors describe how the decision was made to have an 89% credible/confidence interval? I understand that the typical intervals of 90 or 95% are somewhat arbitrary cutoffs but it seems 89% would need to be justified somehow, especially since it seems the ranges generally aren't close to crossing zero (e.g., p. 25).

More minor suggestions:

I thought more information from the supplementary materials would be more useful in the main text. For instance, while reading the main text, I was wondering what the different group types were – perhaps those could be listed in a main text table. This seems important information given the emphasis on the use of novel group types compared to previous studies. Additionally, I thought it would be helpful to include (in the main text) a few examples of the types of items used to measure prestige and dominance, as well as the scale ranges. I was confused by the description of prestige vs. dominance ratings that were approximately 1 point apart but were described as demonstrating perceptions as “highly prestigious but not highly dominant” (p. 25).

Author's Response to Decision Letter for (RSOS-181621.R0)

See Appendix A.

RSOS-181621.R1 (Revision)

Review form: Reviewer 2

Is the manuscript scientifically sound in its present form?

Yes

Are the interpretations and conclusions justified by the results?

Yes

Is the language acceptable?

Yes

Is it clear how to access all supporting data?

Yes

Do you have any ethical concerns with this paper?

No

Have you any concerns about statistical analyses in this paper?

No

Recommendation?

Accept with minor revision (please list in comments)

Comments to the Author(s)

I appreciate the efforts taken by the authors to address some of my prior questions and comments in their response. I also think many of the small edits that were made in the manuscript to

address these points are helpful. I had just a few points of clarification I wanted to raise that I think would further strengthen the paper, which I hope the authors will consider.

It occurred to me that part of my prior confusion about how the authors were discussing prestige and dominance likely stemmed from a lack of clear definition of these terms in the context of the study. I think it would be helpful to explicitly define what the authors mean by prestige and dominance (in addition to the existing discussion about who gains these characteristics and what outcomes they gain from holding these characteristics).

The authors refer to their study as an experiment, but it doesn't seem to me that any variable has been manipulated in the study. I think it would be clearer and more accurate to refrain from describing the study as an experiment.

Finally, I also appreciate that the authors made efforts to anonymise many of the non-public people named by participants in the survey about wider community members. I wanted to ask if the authors could take another look at the "want to learn" list – some names of private citizens are listed with the person's relationship to the participant, and there are some names that seem relatively unusual. Again, I may be unusually careful with respect to these anonymisation concerns, but I think the authors would be practicing admirable open science/data even with some additional non-public persons' names anonymised.

Decision letter (RSOS-181621.R1)

Dear Dr Brand:

On behalf of the Editors, I am pleased to inform you that your Manuscript RSOS-181621.R1 entitled "Prestige and dominance based hierarchies exist in naturally occurring human groups, but are unrelated to task-specific knowledge" has been accepted for publication in Royal Society Open Science subject to minor revision in accordance with the referee suggestions. Please find the referees' comments at the end of this email.

The reviewers and Subject Editor have recommended publication, but also suggest some minor revisions to your manuscript. Therefore, I invite you to respond to the comments and revise your manuscript.

- Ethics statement

- Data accessibility

It is a condition of publication that all supporting data are made available either as supplementary information or preferably in a suitable permanent repository. The data accessibility section should state where the article's supporting data can be accessed. This section should also include details, where possible of where to access other relevant research materials such as statistical tools, protocols, software etc can be accessed. If the data has been deposited in an external repository this section should list the database, accession number and link to the DOI for all data from the article that has been made publicly available. Data sets that have been

deposited in an external repository and have a DOI should also be appropriately cited in the manuscript and included in the reference list.

If you wish to submit your supporting data or code to Dryad (<http://datadryad.org/>), or modify your current submission to dryad, please use the following link:
<http://datadryad.org/submit?journalID=RSOS&manu=RSOS-181621.R1>

- **Competing interests**

- **Authors' contributions**

- **Acknowledgements**

- **Funding statement**

Because the schedule for publication is very tight, it is a condition of publication that you submit the revised version of your manuscript before @@author due date will be populated when the email is sent@@. Please note that the revision deadline will expire at 00.00am on this date. If you do not think you will be able to meet this date please let me know immediately.

When submitting your revised manuscript, you will be able to respond to the comments made by the referees and upload a file "Response to Referees" in "Section 6 - File Upload". You can use this

to document any changes you make to the original manuscript. In order to expedite the processing of the revised manuscript, please be as specific as possible in your response to the referees.

on behalf of Dr Antonia Hamilton (Subject Editor)
openscience@royalsociety.org

Reviewer comments to Author:
Reviewer: 2

Comments to the Author(s)

I appreciate the efforts taken by the authors to address some of my prior questions and comments in their response. I also think many of the small edits that were made in the manuscript to address these points are helpful. I had just a few points of clarification I wanted to raise that I think would further strengthen the paper, which I hope the authors will consider.

It occurred to me that part of my prior confusion about how the authors were discussing prestige and dominance likely stemmed from a lack of clear definition of these terms in the context of the

study. I think it would be helpful to explicitly define what the authors mean by prestige and dominance (in addition to the existing discussion about who gains these characteristics and what outcomes they gain from holding these characteristics).

The authors refer to their study as an experiment, but it doesn't seem to me that any variable has been manipulated in the study. I think it would be clearer and more accurate to refrain from describing the study as an experiment.

Finally, I also appreciate that the authors made efforts to anonymise many of the non-public people named by participants in the survey about wider community members. I wanted to ask if the authors could take another look at the "want to learn" list - some names of private citizens are listed with the person's relationship to the participant, and there are some names that seem relatively unusual. Again, I may be unusually careful with respect to these anonymisation concerns, but I think the authors would be practicing admirable open science/data even with some additional non-public persons' names anonymised.

Author's Response to Decision Letter for (RSOS-181621.R1)

See Appendix B.

Decision letter (RSOS-181621.R2)

05-Apr-2019

Dear Dr Brand,

I am pleased to inform you that your manuscript entitled "Prestige and dominance based hierarchies exist in naturally occurring human groups, but are unrelated to task-specific knowledge" is now accepted for publication in Royal Society Open Science.

on behalf of Prof Antonia Hamilton (Subject Editor)
openscience@royalsociety.org

Appendix A

Editor's comments:

We are awarding a major revision decision to your paper as the reviewers have identified a number of matters that must be addressed prior to the paper being considered further for publication. The reviewers and Editors also note that the anonymisation of study participants is incomplete in the electronic supplements -- please look again at this and ensure A) that you check this with your ethical review board and B) respond to this concern (and how you have tackled it) in the response to referees in your revision. If you need a week or two longer to achieve this owing to the time of year, please let the editorial office know.

Thank you for your response. Regarding the anonymization, we can reassure the editor and reviewers that our study participants were fully anonymised throughout the study as they were given ID numbers for identification and their names were never used or recorded at any time. The names recorded at the end of the supplementary material are those that participants listed as individuals who they deem to be influential members of their wider community (e.g. local MPs or celebrities) and do not include any participants' names.

==REVIEWER COMMENTS==

To the authors:

Here are a few examples (from the supplementary materials) of people who I think are non-public figures whose names were listed:

Aade le Pichon (Mother)
Ben Towe (Head teacher)
Christopher Gray (Director of Music, Cathedral)
Colin Cheesman (Neighbour)
Dr Peter Rodgers (GP/chairman of tennis club)
John Goodman (Newquay Facebook Community Admin)
Lesley Chenels (Day care centre manager)
Monty Pearse (Step father)

Although these names are not the names of participants in the study, I wonder about including the names of such non-public figures (e.g., someone's mother) in a publicly available database for this study. This may be overly cautious on my part, but it seems that it wouldn't be terribly difficult for someone in one of these communities to identify the child of Aade le Pichon, for example. If publishing names of private citizens in the community is not an issue for the local IRB of the authors, I have no special complaints, but I wanted to double check on this issue.

Thank you for highlighting this possibility. We have now removed any identifying information of non-public local figures in the supplementary material, as well as the online data files. We have also removed any link between the lists of community figures (alphabetised) and the participant or group numbers in the online version of the data files, so that, even if someone from the community were to know of a participant or group that took part, they could not possibly trace which community figure that participant/group named. We hope this has now satisfied any concerns.

Comments to Author:

Reviewers' Comments to Author:

Reviewer: 1

Comments to the Author(s)

This is neat little study and a well written paper. I have only a couple of comments that I think could be addressed.

Thank you for your comments and for taking the time to read our manuscript.

(1) l. 647: it is surely doubtful that the only thing people learn in ethnographic societies is manual skills! Knowledge is equally important, not to mention acquiring group-specific cultural knowledge like folktales.

This is true, we were merely stating that the majority of current evidence pertains to learning a skill and that knowledge is a relatively neglected avenue of research. We have adjusted the wording accordingly and hope this is now clearer. (lines 659, 664)

(2) much is made in the Introduction about the fact that this study will look at natural groups rather than groups of strangers (as in the Chen study -- and indeed, most experimental studies in the social sciences). Yet hardly anything is made of this in the Discussion. This is surely the most interesting aspect of the whole study? Almost everything in this area is done with strangers on the apparent assumption that our lives consist in buying secondhand cars from untrustworthy strangers. This might sometimes be the case in economics, but is simply isn't that common in real social life, most of which involves people we know personally and only very occasionally involves strangers. So perhaps a bit more discussion of the differences between the Cheng study and this one in the light of this would be in order? If there are significant differences (as there were in Henrich's tribal economic games studies), this has very important implications for the way we do any experiments in the social sciences.

We agree with the reviewer on the importance of this aspect of our study, hence its prominence in the Introduction. Paragraph 4 of the Discussion also makes this point, and we have now added emphasis to the relevant points in that paragraph (lines 566, 579, 584).

Reviewer: 2

Comments to the Author(s)

This manuscript examines the emergence of prestige and dominance perceptions among small group members in "naturally occurring" human groups. Thirty groups of five members formed on the basis of specific interests (e.g., chess, knitting) were tested individually on general knowledge quizzes, had an opportunity to discuss and submit quiz answers as a team, and then nominated a representative to answer additional quiz questions. All group members also rated each group member, as well as members of the broader community, on dimensions of prestige, dominance, and influence. The authors find that higher ratings on prestige and dominance were associated with preexisting group leaders, but these ratings did not predict who would be nominated as group representative. Rather, the group representative nomination was associated with the highest individual quiz score, although the individual scores were not known.

I read this manuscript with interest, and I appreciated the efforts taken to obtain a diverse community sample of groups with varying sets of interests. I also appreciated how the authors reported both supported and unsupported hypotheses in an unbiased way. I had some questions and comments that I think would help to clarify some aspects of the paper.

Thank you for your kind comments, and for taking time to read the manuscript so thoroughly.

Hypotheses:

It would be helpful if the authors could explain some more about the theorizing regarding the prestige and dominance constructs. I understand that in previous work (Cheng et al.) prestige and dominance were shown to be independent such that “influential group members are either prestigious or dominant, but not both” (p. 12). I also understand that prestige and dominance are separable dimensions, but it is not as clear to me why they should be independent (H1, p. 10). For instance, the authors show in the current data that prestige and dominance ratings are negatively associated. In addition, it seems plausible that there are examples of figures who exhibit both traits, people who have a high level of expertise and a dominating personality, as well as, conversely, people who have low expertise and a more subordinate personality.

We hope that our description of dominance and prestige in the Introduction (lines 49-63) make clear why these two dimensions should be distinct and separable: prestige is based on social learning and skill, such that highly skilled individuals are given prestige in order for others to gain access to them and learn from them, while dominance is based on coercion and fear, such that coercive individuals gain dominance by instilling fear in others. They are distinct because they are generated by different mechanisms. By ‘independence’ we (and Cheng et al.) mean statistical independence. If two measures are independent in a sample, this does not mean that specific individuals cannot be high in both, or low in both; it means that across the sample, there is no statistical relationship between the measures, and other individuals are high in one and low in another. We have clarified this in places (e.g. line 186), and refer the reviewer to the cited Henrich & Gil-White (2001) and Cheng et al. (2013) papers for further details.

I am also a bit confused about some of the examples given for each of these dimensions —why is a teacher an example of a dominant/high status person as opposed to a person from whom to learn (p. 18)?

The reference to a teacher is asking participants to name “influential” members of their group. Influential is used throughout our study to refer to someone who could *either* be prestigious *or* dominant, as previous work suggests that prestige and dominance are two distinct routes to gaining influence over a group (see previous point). Thus, a kayak club may have a teacher who is influential within the group, and this teacher may then be rated as either prestigious or dominant, or a bit of both (see previous point). We might expect that teachers are influential because of their expertise, and hence their prestige, and not their dominance; this was exactly what we set out to investigate in our study.

Almost all the predictions are about prestige, and it seems the only prediction regarding dominance is about overconfidence. This left me wondering what the utility of examining dominance is. Theoretically, why do group members tolerate dominant members? Do dominant members serve some purpose, such as being perceived as providing more existential safety from outgroup threats compared to prestigious members (who perhaps serve more epistemic needs)? Or are dominant members tolerated simply because they are feared and might otherwise serve an existential threat?

The main focus of our predictions were indeed about prestige, as the theory we are testing is predominantly about the purported evolution of a unique prestige hierarchy in humans (as opposed to dominance hierarchies that are also found in other species). Thus, we are more interested in testing if and when prestige hierarchies emerge in humans as opposed to (or as well as) dominance hierarchies. In particular, we were interested in the relationship that prestige has to knowledge-based domains, as we predict prestige should be more favoured here (and dominance less so). There are interesting theories relating to when people may prefer dominant leaders, e.g. when going to war, but this was not a focus of our current study.

Relatedly, I wondered whether there could be a public/private dimension that affects the nominations. That is, would nominations that are made publicly (i.e., in front of the group) be driven by dominance ratings?

This is an interesting point and we debated whether the voting should be public or private. We decided on private as we reasoned that voting order in a public vote could influence subsequent voters, and controlling for this would be practically very tricky. A public vote would also answer a slightly different question, (i.e. how do people vote when they know the group is watching/listening, or how are peoples' votes affected by others' votes). We chose to implement a private vote as we wanted to get as honest an opinion as possible, hopefully revealing peoples' genuine preferences based on the group interaction. This is now mentioned on lines 179-181.

Theory/Design:

As mentioned earlier, I like that the authors examined existing interest-based groups for greater ecological validity than arbitrarily assigned student groups. However, the advantages of examining the existing groups seem somewhat lost with the current design. That is, it's not clear to me what we learn about naturally occurring groups if they are essentially treated as arbitrary groups. I would have thought that it is potentially more useful to understand the emergence and existence of prestige and dominance perceptions within the context of each group's interest. For example, for a group formed on the basis of interest in chess, it seems that assessments of prestige, dominance, influence, and learning opportunities would be best assessed on the topic of chess.

We weren't initially interested in the idiosyncrasies of the individual groups- as is shown by our preregistered predictions, we were expecting to replicate previous studies with unacquainted groups, so predicted their prestige ratings to be related to their performance on the quiz. However, this prediction was not upheld, and we found an unexpected finding that prestige was tied to existing positions within the groups. This was not something we were initially looking to investigate, therefore we haven't emphasised between-group differences in this study. We agree this would be an interesting avenue for future research. Our prestige, dominance, influence and learning questionnaires were not tied to the quiz, but were general questions about the participants' general tendencies, and so could have been related to chess within the chess teams (and probably were). The only ratings related to the quiz were the group votes, which were not related to the groups' previous interactions, but on the quiz itself. This is now discussed explicitly in the Discussion, lines 588-594.

As the authors acknowledge, it appears that participants' judgments were based on preexisting attitudes formed in the context of the group's organizing interest and prior interactions, and it seems such attitudes are stronger than the immediate study context of general knowledge quizzes. Presumably group members don't know how generally knowledgeable each other are until they have the group discussion, which makes them functionally similar to the arbitrary college groups in prior work. This point is touched upon in the discussion, but it seems a more important consideration for the paper's theorizing to me.

It may be the case that groups didn't know about each other's general knowledge until the group discussion, but we don't know that for sure. Even if this did make them functionally similar to arbitrary groups, our results show that they weren't functionally similar, in that they based their prestige/dominance ratings on previous interactions, rather than the general knowledge discussion. We have emphasised this more in the paragraph of the discussion which touches on this (lines 550-557 and 564-594).

As the authors note, the broader hypotheses about prestige are not supported in this study in that higher prestige ratings in the group do not translate to higher perceptions of potential learning, likely because prestige ratings seem to be based on preexisting attitudes, whereas the social learning is specific to the general knowledge quiz. It would be helpful if the authors could elaborate on the potential short-term vs. long-term or context-specific relationship between prestige and social learning.

This is a good point and is nicely addressed in Redhead et al's recent paper, which we have now cited in our discussion, line 587.

Analyses:

I have a few small comments/suggestions that I think make the analyses easier to understand for readers who may not have expertise with multilevel models (for example, me):

I think the analyses section could be more helpful if it were integrated into the results. As it is now, the analysis section reads as a list of separate decisions that aren't so easy to follow because it's not always obvious what they are for (e.g., on p. 23 it is stated that "models were said to be a better fit to the data if they had WAIC value..." but I think the model has not yet been specified. What are the variables being accounted for? In fact, I don't think I saw any section in which the model was clearly specified. For instance, was dominance regressed on prestige? What are other covariates being used?).

Thank you for highlighting this, we have now altered the analysis section to hopefully be clearer, starting on line 337. We have highlighted that Tables 1, 2, and 3 include all of the model parameters including the varying effects and predicted outcomes, and have altered the table legends to emphasise this.

In general, I think it would be helpful to get more explanation about the analysis decisions. For example, could the authors describe how the decision was made to have an 89% credible/confidence interval? I understand that the typical intervals of 90 or 95% are somewhat arbitrary cutoffs but it seems 89% would need to be justified somehow, especially since it seems the ranges generally aren't close to crossing zero (e.g., p. 25).

We have highlighted that 89% intervals are default in the Rethinking package, and that using 95% intervals, as is the arbitrary norm, would not alter the interpretation of our results.

More minor suggestions:

I thought more information from the supplementary materials would be more useful in the main text. For instance, while reading the main text, I was wondering what the different group types were — perhaps those could be listed in a main text table. This seems important information given the emphasis on the use of novel group types compared to previous studies. Additionally, I thought it would be helpful to include (in the main text) a few examples of the types of items used to measure prestige and dominance, as well as the scale ranges.

As our manuscript is already heavily loaded with tables and figures of results and model specifications, we would rather keep additional information that is not relevant for our predictions, analyses or interpretations in the supplementary material. As between-group differences were not a

measure of interest in this particular study, we would rather keep the main manuscript free from distractions. Readers who are particularly interested in our study groups can access that information not only in the supplementary material but also on the online Github repository listed in the text.

I was confused by the description of prestige vs. dominance ratings that were approximately 1 point apart but were described as demonstrating perceptions as “highly prestigious but not highly dominant” (p. 25).

Although it appears these ratings are approximately only 1 point apart, these are ordinal Likert scale ratings (1-7), and so a difference of one point on the scale can be quite meaningful. Nevertheless, we have altered the wording so as not to be confusing/misleading (line 394).

Appendix B

Reviewer comments to Author:

Reviewer: 2

Comments to the Author(s)

I appreciate the efforts taken by the authors to address some of my prior questions and comments in their response. I also think many of the small edits that were made in the manuscript to address these points are helpful. I had just a few points of clarification I wanted to raise that I think would further strengthen the paper, which I hope the authors will consider.

It occurred to me that part of my prior confusion about how the authors were discussing prestige and dominance likely stemmed from a lack of clear definition of these terms in the context of the study. I think it would be helpful to explicitly define what the authors mean by prestige and dominance (in addition to the existing discussion about who gains these characteristics and what outcomes they gain from holding these characteristics).

Thank you for highlighting this, we have now adjusted the wording in the abstract and the introduction to be clearer about how prestige and dominance are defined in our study.

The authors refer to their study as an experiment, but it doesn't seem to me that any variable has been manipulated in the study. I think it would be clearer and more accurate to refrain from describing the study as an experiment.

Whilst we acknowledge that no experimental manipulations were carried out, we feel the research is still best described as an experiment rather than purely observational, as each group was tested in controlled conditions with identical measures taken across all individuals and across all groups.

Finally, I also appreciate that the authors made efforts to anonymise many of the non-public people named by participants in the survey about wider community members. I wanted to ask if the authors could take another look at the "want to learn" list – some names of private citizens are listed with the person's relationship to the participant, and there are some names that seem relatively unusual. Again, I may be unusually careful with respect to these anonymisation concerns, but I think the authors would be practicing admirable open science/data even with some additional non-public persons' names anonymised.

Although some names listed may be those of private citizens, we as authors have no way of knowing which names represent public figures, or semi-public (i.e. locally well-known) figures unless we research every name independently. For example, many of those names listed are locally-influential artists. Thus, we either have to anonymise every entry (including obvious public figures such as Jeremy Corbyn/David Attenborough etc) or none of the entries. Although the "commRatings" included information about the individuals' position, and thus could potentially be linked to a particular group, this list has no such information as it only lists traits the others would like to learn from them. None of the names can be associated with any of the participants, even those named as 'mother' as they are decoupled from the rest of the database and reordered, and given that there is no way of knowing which participant listed which individual, we feel it is appropriate to leave the data as they are as they do not breach our ethics approval in any way.